# The Antibacterial Efficacy and In Vivo Toxicity of Sodium Hypochlorite and Electrolyzed Oxidizing (EO) Water-Based Endodontic Irrigating Solutions

**DOI:** 10.3390/ma13020260

**Published:** 2020-01-07

**Authors:** Sung-Chih Hsieh, Nai-Chia Teng, Chia Chun Chu, You-Tai Chu, Chung-He Chen, Liang-Yu Chang, Chieh-Yun Hsu, Ching-Shuan Huang, Grace Ying-Wen Hsiao, Jen-Chang Yang

**Affiliations:** 1School of Dentistry, Taipei Medical University, Taipei 110-52, Taiwan; endo@tmu.edu.tw (S.-C.H.); tengnaichia@hotmail.com (N.-C.T.); jollyhuangtw12@gmail.com (C.-S.H.); gracehsiaodds@gmail.com (G.Y.-W.H.); 2Graduate Institute of Nanomedicine and Medical Engineering, College of Biomedical Engineering, Taipei Medical University, Taipei 110-52, Taiwan; zhy5eqf0i@gmail.com (C.C.C.); chudaev1977@ukr.net (Y.-T.C.); gpchaucer@gmail.com (C.-H.C.); alex627324@gmail.com (L.-Y.C.); alice1226@tmu.edu.tw (C.-Y.H.); 3Research Center of Biomedical Device, Taipei Medical University, Taipei 110-52, Taiwan; 4International Ph.D. Program in Biomedical Engineering, College of Biomedical Engineering, Taipei Medical University, Taipei 110-52, Taiwan; 5Research Center of Digital Oral Science and Technology, Taipei Medical University, Taipei 110-52, Taiwan

**Keywords:** irrigation solutions, vital pulp therapy, zebrafish, *Streptococcus mutans*, *Enterococcus faecalis*

## Abstract

The objective of this study was to evaluate the antibacterial efficacy against *Enterococcus faecalis* and *Streptococcus mutans* and in vivo toxicity using embryonic zebrafish assays of sodium hypochlorite (NaOCl) and electrolyzed oxidizing (EO) water (containing hypochlorous acid (HOCl))-based root canal irrigating solutions. Methodology: Using 100 μL microbial count of 1 × 10^8^ cfu/mL *Enterococcus faecalis* to mix with each 10 mL specimen of NaOCl or HOCl for designed time periods. The above protocol was also repeated for *Streptococcus mutans*. The concentration of viable microorganisms was estimated based on each standardized inoculum using a plate-count method. Zebrafish embryo assays were used to evaluate acute toxicity. Results: All the HOCl or NaOCl treatment groups showed > 99.9% antibacterial efficacy against *Enterococcus faecalis* and *Streptococcus mutans*. Zebrafish embryos showed almost complete dissolution in 1.5% NaOCl within 5 min. Both survival rates after being treated with 0.0125% and 0.0250% HOCl for 0.5 min or 1.0 min were similar to that of E3 medium. Conclusions: Both NaOCl and HOCl revealed similar antibacterial efficacy (> 99.9%) against *Enterococcus faecalis* and *Streptococcus mutans*. While 1.5% NaOCl fully dissolved the Zebrafish embryos, both 0.0125% and 0.0250% HOCl showed little in vivo toxicity, affirming its potential as an alternative irrigation solution for vital pulp therapy.

## 1. Introduction

The removal of the pathologic pulp, cleaning, and shaping of the root canal system are essential for successful endodontic treatment to disinfect pulpal space or prevent its reinfection of reference [1]. The complex root anatomy is one of the major challenges to intracanal cleaning and disinfection; therefore, mechanical instrumentation combined with chemical irrigation is the widely accepted strategy to enhance successful debridement [2]. An ideal irrigant should be effective in removing necrotic pulp tissues, smear layers, and disinfecting dentinal tubules [3].

Among the current available endodontic irrigating solutions, sodium hypochlorite (NaOCl) is known for its ability to digest organic tissues during chemomechanical debridement of the root canals [4]. However, NaOCl’s tissue-dissolving ability is accompanied by its cytotoxicity to vital tissue [5]. It has been documented that severe adverse reactions such as sequelae of pain, swelling, ecchymosis, and paresthesia can occur as a result of chemical burn associated with NaOCl periradicular extravasation during endodontic treatment [6]. For this reason, AAE (American Association of Endodontists) recommends use of low concentrations of NaOCl (1.5% for 5 min) followed by irrigation with saline or ethylenediaminetetraacetic acid (EDTA) to minimize cytotoxicity to stem cells in regenerative procedure [7]. Additionally, hypersensitivity and allergic reactions to NaOCl have been reported, and it has been suggested that when allergy and hypersensitivity are verified, endodontic treatment needs to be carried out with alternative irrigants [8]. Many studies have also proposed the use of two or more endodontic irrigants to target different bacterial spectrum in order to increase treatment success, but there are limitations when NaOCl is used in conjunct with other effective endodontic irrigants such as EDTA and chlorohexidine. When NaOCl is combined with EDTA, NaOCl loses its tissue-dissolving capacity and antibacterial efficacy because virtually no free chlorine was detected in the combination [9]. Likewise, if NaOCl is added to chlorhexidine, it may have a synergistic effect initially. However, this does not increase its long term (residual) antibacterial activity [10], and a precipitate forms when these two agents are mixed that can present the risk of discoloration and potential leaching of unidentified chemicals into the periradicular tissues [11]. All of the above side effects with the use of NaOCl have prompted our search for an alternative endodontic irrigant that has superior antimicrobial capability along with little tissue toxicity.

The electrolyzed oxidizing (EO) water can be produced by the electrolysis of a dilute NaCl aqueous solution through an electrolysis chamber. Similar to NaOCl as a chlorine-based sanitizer, EO water comprised of hypochlorous acid (HOCl) usually possesses an extraordinary bactericidal effect due to its high reduction/oxidation potential (ORP) over 1100 mV [12]. Whereas HOCl carries no electrical charge, hypochlorite ion on the contrary carries a negative electrical charge, resulting in a repulsion of the germs with negatively charged cell wall surfaces, making hypochlorite ion less effective at killing germs [13].

Besides its usage as a standard irrigant in endodontic procedures, NaOCl is commonly used in procedures trying to preserve pulp’s vitality, like in the case of deep carious lesions where the pulp could be inflammed but is not infected. *Streptococcus mutans* and *Enterococcus faecalis* were the bacteria of choice in this study due to their prevalence in caries (*S. mutans*) and in teeth needing endodontic retreatment (*E. faecalis)* [14].

Among piscine models, the zebrafish is a popular vertebrate model to study the toxicity of various xenobiotic agents [15]. No license is required if the zebrafish embryos used are less than 5 days old under the European Union, Directive 2010/63/EU (revised from Directive 86/609/EEC) [16].

Vital pulp therapy is a well-defined treatment initiated to preserve and maintain pulp tissue in a healthy state [17]. Vital pulp therapy typically starts with decay removal with subsequent stepwise excavation of the inflamed pulpal tissue; treatment includes indirect pulp capping, direct pulp capping, miniature pulpotomy, partial/Cvek pulpotomy, and coronal/complete pulpotomy [18]. Maintaining the vitality of the dental pulp increases a tooth’s mechanical resistance and long-term survival. A root canal irrigant solution with balance between cytotoxicity and antibacterial properties is essential for successful disinfection and cleanness in endodontic procedures. Our purpose is to investigate the antibacterial property and cytotoxicity of EO water containing HOCl, relative to NaOCl.

## 2. Materials and Methods

### 2.1. Preparation of Electrolyzed Oxidizing (EO) Water

Unless otherwise specified, all chemicals were purchased from Sigma-Aldrich (St. Louis, MO, USA) and used without further purification. The ANK-Neutral Anolyte GH-40 (Envirolyte Industries International Ltd., Tallinn, Estonia) was used to produce 50 L/h EO water by mixing DD water with an over-saturated solution of sodium chloride under 110 V. The output concentration of (HOCl) was set at the range of 330–350 ppm. The residual chlorine concentration of the EO water was measured using a HI96771 chlorine photometer (Hanna Instruments, Woonsocket, RI, USA) first, then diluted to 0.0125% and 0.0250%, respectively.

### 2.2. Antibacterial Efficacy

Prior to the test, *Enterococcus faecalis* (*E. faecalis*, ATCC^®^ 29212) cultures were inoculated on the surface of tryptic soya agar with polysorbate 80, lecithin (Sigma-Aldrich/51414, St. Louis, MO, USA), and 5% defibrinated sheep blood at 37 °C for 48–72 h, then adjusted microbial count to about 10^8^ colony-forming units (cfu)/mL. Same method applied to obtaining *Streptococcus mutans* (*S. mutans,* ATCC^®^ 25175) cultures. One hundred microliters microbial count of 1 × 10^8^ cfu/mL *E. faecalis* and *S. mutans,* respectively, was used to mix with each 10 mL specimen of NaOCl (1.5% and 5.25%) and HOCl (0.0125% and 0.0250%) to give an inoculum of 10^5^ to 10^6^ cfu/mL for designed contact time periods. A suitable sample was removed immediately from each suspension and the number of cfu/mL in each suspension was determined by plate-count method according to United States Pharmacopeia (USP) Chapter < 51 > antimicrobial effectiveness testing.

### 2.3. Zebrafish Embryo Toxicity Assays

The animal ethic approval (LAC-2019-0243) from Taipei Medical University ethics committee was obtained. One hundred fifty fertilized wild-type zebrafish (Danio rerio) eggs 1-h post fertilization (1 hpf) were moved to Petri dishes and incubated within the zebrafish embryo E3 medium (5 mM NaCl, 0.17 mM KCl, 0.33 mM CaCl_2_, and 0.33 mM MgSO_4_) at a temperature of 28 °C. To evaluate acute toxicity and developmental defects caused by NaOCl and EO water, zebrafish embryos were exposed to sodium hypochlorite (NaOCl) (1.5%) for 5 min as well as EO waters (0.0125% and 0.0250%) for 0.5 and 1.0 min using a yellow 100 µm cell strainer (Falcon^®^, One Becton Circle, Durham, NC, USA), then transferred to Petri dishes (*n* = 10) and three replications and recorded data at representative stages (24, 48, and 72 hpf). The survivability and conditions of the embryos were captured under a light microscope (Olympus SZX16, Shinjuku-ku, Tokyo, Japan) and a digital camera (Canon EOS 550D, Ohta-ku, Tokyo, Japan) under 40× and 100× magnifications. Percentage survival of the embryos was evaluated and scored for lethal or teratogenic effects.

### 2.4. Statistical Analysis

One-way analysis of variance (ANOVA) was used to evaluate the statistical significance of the measured data. Duncan’s test comparison was used to determine the significance of deviations (*p* < 0.05) in the measured data of each group.

## 3. Results

### 3.1. Antibacterial Properties for HOCl and NaOCl

The bacterial counts and reduction of *E. faecalis* and *S. mutans* before and after treatment are summarized in Table 1, accordingly. All the HOCl or NaOCl treatment groups showed over a 5 log 10 cfu/mL reduction in *E. faecalis* and *S. mutans* population, indicating > 99.9% antibacterial efficacy.

### 3.2. Zebrafish Embryonic Toxicity Test

The mortality of Zebrafish embryos exposed to different concentrations of HOCl and NaOCl was determined at specific time points. Figure 1 shows photomicrographs of zebrafish embryo under conditions of positive control (1.5% NaOCl) and negative control (E3 medium), in 0.0125% HOCl for 0.5 min and in 0.0250% HOCl for 0.5 min, respectively. Zebrafish embryo was almost fully dissolved in 1.5% NaOCl within 5 min; while zebrafish embryos hatched and grew healthily in E3 medium up to 72 phf.

The survival rates of Zebrafish embryos exposed to various concentrations of HOCl or NaOCl at different soaking times were investigated. Table 2 showed the survival rates at 72 phf for zebrafish embryos after exposure to 0.0125% HOCl, 0.0250% HOCl, and E3 medium. Except for the percent survival rate for the group of 0.0250% HOCl with 1.0 min and E3 medium, there were no significant differences among the rest of the groups (*p* > 0.05).

## 4. Discussion

Vital pulp therapy is aimed to remove source of infection from dental caries during early stages of the inflammation, thus healthy pulp tissue can be preserved to prevent complete necrosis. Various degrees of vital pulp therapy are available, such as indirect pulp capping, direct pulp capping, or pulpotomy, which all depend on stringent aseptic techniques to be successful [19]. NaOCl is known as a long-accepted endodontic irrigant due to its excellent antimicrobial property; its mechanism includes biosynthetic alteration in cellular metabolism and phospholipid destruction, the formation of chloramines (which interferes in cellular metabolism), an oxidative action with irreversible enzymatic inactivation in bacteria, and a lipid and fatty acid degradation [20]. In regard to vital pulp therapy, NaOCl is recommended as a part of standard protocol in Carisolv chemomechanical caries removal system and partial pulpotomy by Cvek. Although NaOCl has been the irrigant of choice, it needs to be noted that low concentration is used in Carisolv gel [21] and Cvek partial pulpotomy [22] to prevent tissue toxicity. In the present study, bactericidal effect of NaOCl and EO waters containing HOCl was evaluated against *S. mutans* and *E. faecalis*. Results from Table 1 showed > 99.9% antibacterial efficacy against *S. mutans* and *E. faecalis* in all the NaOCl and HOCl treatment groups, suggesting low concentration such as 0.0125% and 0.0250% HOCl is equally effective as 1.5% and 5.25% NaOCl. *S. mutans* is one of the prevalent species found in complex microflora of dental caries [23], and study has shown that reducing population of *S. mutans* in oral infection greatly reduces dental caries in vivo [24]. Due to similar antibacterial efficacy demonstrated in our study, EO water containing low concentration of HOCl can be substituted in place of NaOCl in vital pulp therapy when the predominant bacteria are *S. mutans*. While *S. mutans* plays a major role in dental decay that may require vital pulp therapy depending on the amount of tooth structure damage, *E. faecalis* on the other hand is a common persistent pathogen in endodontic infections and the dominant species recovered in failed endodontic cases [14]. Various studies have been reported to be effective but to a different extent regarding antimicrobial properties, depending on *E. faecalis* phenotype and duration of contact with the agent [25]. In other studies, reported proportion of dead *E. faecalis* cells in the treatment of 5.25% NaOCl group for 10 min was 94.14% for 3-week-old biofilm model [26] but 96.3% for continuous fluid flow biofilms [27]. From the results of this study, HOCl is potent in killing *E.* faecalis in low concentration, thus making it a potential replacement for NaOCl as irrigating solution in failed endodontic treatment.

Several studies have depicted antimicrobial mechanism of HOCl as an innate substance produced to fight infection. Similar to intrinsic HOCl produced from the human body’s immune cells by the myeloperoxidase-H_2_O_2_-Cl system of phagocytes [28], EO water containing HOCl is as capable in fighting invading pathogens and infections [29]. General mechanism for the bacterial toxicity of HOCl was reported by abolition of adenosine triphosphate (ATP) production [30]. The 0.0125% and 0.0250% HOCl revealed similar antibacterial efficacy (> 99.9%) against *E. faecalis* and *S. mutans*, when compared with 5.25% NaOCl. Similar outcomes for using HOCl as an endodontic irrigating solution for root canal cleanliness and smear layer removal in the root canals of ex vivo human teeth were verified [31]. In addition, the cytotoxicity of HOCl against pulp cells was mild compared to that of NaOCl solution [32], and this tissue-friendly effect was also confirmed from our study as shown in Figure 1c,d when zebrafish embryos treated with 0.0125% HOCl and 0.0250% HOCl were able to hatch and grow to healthy size when compared with negative control. EO waters containing low concentration of HOCl are a promising alternative to NaOCl in vital pulp therapy when the aim of such treatment is to eradicate infection but also to preserve healthy pulp tissue to promote further root formation. By substituting irrigating solution with EO water, cytotoxicity of NaOCl or NaOCl/CHX [33] is eliminated in vital pulp therapy.

Current research in regenerative endodontics uses mainly improved materials, instruments, and medications [34]. The concept of regenerative endodontics is to physiologically replace damaged dentin and root structures as well as the pulp-dentin complex [3]. Pulp tissue is vital to dentinogenesis [35], tooth nutrition [36], proprioceptor cognizance [14], and immune cell defense [37]. The preservation of dental pulp is crucial to the long-term function of teeth to stimulate the formation of reparative dentin to retain teeth as functional units. Complete regeneration of the pulp-dentin complex is possible when 1–4 mm of residual dental pulp tissue remains in the apical segment of the root canal system [38].

Grossman and Meiman [39] reported that 5% sodium hypochlorite dissolves pulp tissue between 20 min and 2 h. The dissolution of bovine pulp tissue by NaOCl (0.5%, 1.0%, 2.5%, and 5.0%) was studied in vitro under different conditions [40]. Estrela et al. [41] described the tissue dissolving mechanism of NaOCl associated with (a) saponification reaction to degrade fatty acids and (b) chloramination reaction to release chlorine, which form chloramines in reaction with proteins. Even with strong tissue dissolving capability, the NaOCl cannot remove the smear layer effectively [42]. This tissue-dissolving property was confirmed by our observation of the nearly full dissolution of zebrafish embryo in 1.5% NaOCl within 5 min. The major cell populations found in the pulp include fibroblasts, undifferentiated mesenchymal cells, odontoblasts, macrophages, and other immunocompetent cells. These cells are potentially vulnerable to dissolution during irrigation with 1.5% NaOCl in vital pulp therapy.

One could argue that when cytotoxicity of pulpal tissue is not of concern in cases of pulpectomy of necrotic pulp or when cleaning endodontic space from previous endodontic-treated teeth with persistent infection, NaOCl can continue to serve as the standard irrigant. However, severe complications can occur when NaOCl comes into contact with soft tissue if it is extruded out of apical foramen. These complications may include but are not limited to chemical burns leading to tissue necrosis, tissue swelling that may be oedematous or hemorrhagic, or both due to severe acute inflammatory reaction [43]. Paraesthesia and anaesthesia affecting mental inferior dental and infraorbital branches of the trigeminal nerve have been described, and normal sensation could take several months to regain [44]. Therefore, even when vital pulp tissue is not present, using an antimicrobial but tissue-friendly irrigant should still be preferred to increase greater safety in endodontic treatment.

The concept of “minimally invasive endodontics (MIE)” has been practically adapted by passive ultrasonic irrigation [45], contracted endodontic cavities [46], and self-adjusting file (SAF) system [47] to increase treatment efficacy while preserving structure integrity [48]. More studies are needed to verify the extension of the MIE concept, when using HOCl as irrigating solution to reduce damage to pulp tissue while maintaining successful disinfection and cleanliness in vital pulp therapy. In this study, the efficacy of the irrigants was carried out with individual planktonic bacterial culture; results may differ in in vivo conditions when infected endodontic space is filled with multispecies biofilm. Prior to clinical application, further research should be designed using ex vivo pulp cells to evaluate cytotoxicity, and simulated biofilm tooth model should be used to study antibacterial efficacy of the EO waters containing HOCl.

## 5. Conclusions

Within the limitations of this study, it can be concluded that both EO waters containing 0.0125% and 0.0250% HOCl revealed a remarkable but similar bactericidal effect (> 99.9%) to that of conventional NaOCl against *E. faecalis* and *S. mutans*. Unlike the Zebrafish embryo fully dissolved in 1.5% NaOCl, both 0.0125% and 0.0250% HOCl showed similar survival rate to E3 medium with little or mild in vivo toxicity, revealing the potential as an alternative irrigation solution for vital pulp therapy.

## Figures and Tables

**Figure 1 materials-13-00260-f001:**
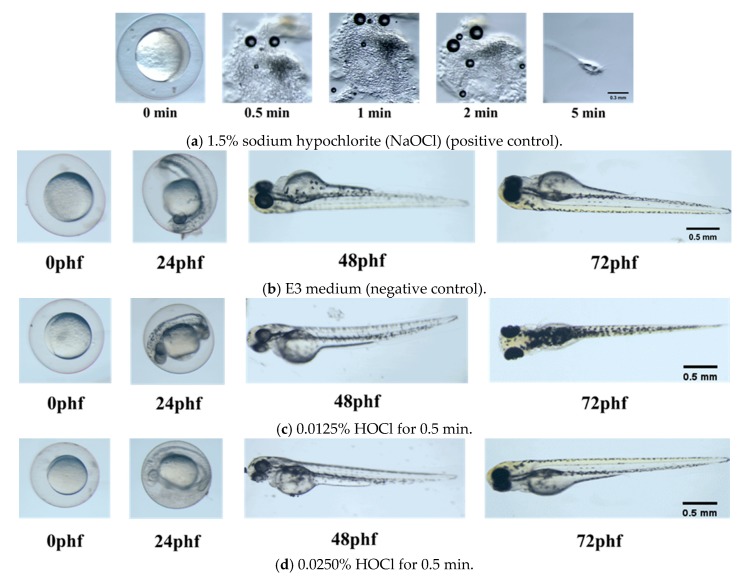
Photomicrograph of Zebrafish embryo. (**a**) Zebrafish embryo in 1.5% NaOCl (positive control). (**b**) Zebrafish embryo in E3 medium (negative control). (**c**) Zebrafish embryo in 0.0125% HOCl for 0.5 min. (**d**) Zebrafish embryo in 0.0250% HOCl for 0.5 min.

**Table 1 materials-13-00260-t001:** Bacterial counts and reduction of (**a**) *E. faecalis* and (**b**) *Streptococcus mutans* before and after treatment.

(**a**) *Enterococcus Faecalis* (ATCC^®^ 29212)				
**Samples**	**Treatment Time** **(min)**	**Before** **(cfu/mL)**	**After** **(cfu/mL)**	**Bacterial Reduction** **(%)**
1.50% NaOCl	5.0	3.8 × 10^5^	<1	>99.9
5.25% NaOCl	0.5	3.8 × 10^5^	<1	>99.9
5.25% NaOCl	1.0	3.8 × 10^5^	<1	>99.9
0.0250% HOCl	0.5	3.8 × 10^5^	<1	>99.9
0.0250% HOCl	1.0	3.8 × 10^5^	<1	>99.9
0.0125% HOCl	0.5	3.8 × 10^5^	<1	>99.9
0.0125% HOCl	1.0	3.8 × 10^5^	<1	>99.9
(**b**) *Streptococcus Mutans* (ATCC^®^ 25175)				
**Samples**	**Treatment Time** **(min)**	**Before** **(cfu/mL)**	**After** **(cfu/mL)**	**Bacterial Reduction** **(%)**
0.0250% HOCl	0.5	4.3 × 10^5^	<1	>99.9
0.0250% HOCl	1.0	4.3 × 10^5^	<1	>99.9
0.0125% HOCl	0.5	4.3 × 10^5^	<1	>99.9
0.0125% HOCl	1.0	4.3 × 10^5^	<1	>99.9

**Table 2 materials-13-00260-t002:** Percent survival rate at 72 h of Zebrafish embryos after exposure to E3 medium, 0.0125%, and 0.0250% HOCl.

Samples	0.5 min Soaking Time		1 min Soaking Time	
	Survival Rate (%) at 72 h	Body Length (mm) at 72 h	Survival Rate (%) at 72 h	Body Length (mm) at 72 h
E3 Medium	83.3 ± 5.8 ^a^	3.55 ± 0.19 ^b^	83.3 ± 5.8 ^a^	3.57 ± 0.22 ^b^
0.0125% HOCl	73.3 ± 5.8 ^a^	3.45 ± 0.18 ^b^	76.7 ± 15.3 ^a,c^	3.22 ± 0.18 ^b^
0.0250% HOCl	66.7 ± 25.2 ^a^	3.41 ± 0.19 ^b^	50.0 ± 17.3 ^c^	3.23 ± 0.36 ^b^
1.5% NaOCl	0	N.A.	0	N.A.

Survival rate (*n* = 3); body length (*n* = 30). Values are the mean ± standard deviation. Mean values followed by the same superscript letter do not significantly differ (*p* > 0.05) according to Duncan’s multiple comparisons. N.A.: not available.

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
