# Peer review of "The Antibacterial Efficacy and In Vivo Toxicity of Sodium Hypochlorite and Electrolyzed Oxidizing (EO) Water-Based Endodontic Irrigating Solutions"

_materials, 2020, doi:10.3390/ma13020260_

Round 1
Reviewer 1 Report
A major english test check is required.
Introduction (line 52): add to the limits of Sodium Hypochlorite the interferences with other irrigants (e.g. EDTA, and Chlorhexidine) Reference: Mohammadi Z, Giardino L, Palazzi F, Shalavi S, Alikhani MY, Lo Giudice G, Davoodpour N. Effect of sodium hypochlorite on the substantivity of chlorhexidine. Int J Clin Dent 2013;6(2):173-178.
Discussion: Clarify the advantages connected to the use of HOCl, underlining limits and indications in endodontic procedures.
Reviewer 2 Report
The article is well written and of scientifical interest.
The use of a less dangerous product for the dysinfection of root canals is an important topic in the field of endodontics. Sodium hypoclorite is irritant for tissues and many cases of extrusion with post-treatment injuries are reported in literature.
References do not respect journal guidelines: please check instructions for authors at: https://mdpi-res.com/data/mdpi_references_guide_v5.pdf
Reviewer 3 Report
INTRODUCTION
Line 43 44 Pathosis pulpal/periraducular cannot be re(infected). It is the result of the infection of endodontic space (pulpal space).
GENERALLY INTRODUCTION The rationale of the study should be clear.
The reason for choosing the microorganisms. Which microorganism is related to caries and which to refractory and persistent infection of endodontic space? This should be mentioned in the introduction as well.
INTRODUCTION AND DISCUSSION
Although in the Discussion it was somewhat better discerned what reffers to vital pulp therapy and what to pulpectomy, it is generally not clear from the introduction and discussion that in vital pulp therapy endodontic space is sterile and that the success of vital pulp therapy depends on the absence/presence of microorganisms in dentin cavity. Iti s not mentioned that the therapy always begins with caries removal. Classical In vitro studies showed that injured pulp tissue did not heal if contaminated, whereas the healing occurred if the pulp tissue had not been infected after exposure/injury. The procedures including vital pulp therapy should be listed (indirect pulp therapy, direct pulp therapy and pulpotomy) and those where the usage of NaOCl is recommended as a part of standard protocol should be mentioned (including chemomechanical approach to caries excavation such as Carisolv system, and treatment with NaOCl after partial pulpotomy by Cvek).
As far as pulpectomy of necrotic pulp and disinfection of infected endodontic space is concerned, NaOCl is a standard irrigant. Its cytotoxicity is not of concern if its action is limited to endodontic space, however if extruded through the apical foramen, could result in serious clinical conditions. This would justify this research as far as the treatment of necrotic pulp is concerned. Cytotoxic effect on pulpal cells is of no interest in such cases, as they are necrotic.
It should however be mentioned that the efficacy of the agents used in this research on individual bacterial cultures, does not necessarily mean that it would be as effective in in vivo conditions where biofilms are formed.
MATERIALS
It would be suitable if the concentrations of both irrigans were expressed in the same units (ppm or %).
Round 2
Reviewer 3 Report
Generally, it should be clear that preserving vitality of the pulp is not endodontic treatment per se.
Please, make minor changes in the inserted paragraphs of Your Introduction in this regard. The rephrased paragraphs are below.
Ad1. The removal of the pathologic pulp, cleaning, and shaping of the root canal system are essential for SUCCESSFULL endodontic treatment to DISINFECT pulpal space OR PREVENT ITS REINFECTION [1].
Ad2. BESIDES ITS USAGE AS A STANDARD IRRIGANT IN ENDOODNTIC PROCEDURES, NaOCl IS COMMONLY USED IN THE PROCEDURES TRYING TO PRESERVE PULP'S VITALITY, LIKE IN THE CASE OF DEEP CARIOUS LESIONS WHERE THE PULP COULD BE INFLAMMED BUT IS NOT INFECTED. Streptococcus mutans and Enterococcus faecalis WERE the bacteria of choice in this study due to their prevalence IN CARIES (S.MUTANS) AND in teeth needing endodontic REtreatment (E.FAECALIS) [14].
Author Response
Very appreciate for this kind instruction. Revisions were made as noted in red.